# Modelling transcription with explainable AI uncovers context-specific epigenetic gene regulation at promoters and gene bodies

**Kashyap Chhatbar**[1,2*], **Adrian Bird**[2], **Guido Sanguinetti**[3]

**1** School of Informatics, University of Edinburgh, Edinburgh, United Kingdom, **2** Institute of Cell Biology, University of Edinburgh, Edinburgh, United Kingdom, **3** International School for Advanced Studies (SISSA), Trieste, Italy

* kashyap.c@ed.ac.uk

**Data availability statement:** The raw sequencing data used in this study were obtained from the Gene Expression Omnibus (GEO) and are publicly available under the

## Abstract

Transcriptional regulation involves complex interactions with chromatin-associated proteins, but disentangling these mechanistically remains challenging. Here, we generate deep learning models to predict RNA Pol-II occupancy from chromatin-associated protein profiles in unperturbed conditions. We evaluate the suitability of Shapley Additive Explanations (SHAP), a widely used explainable AI (XAI) approach, to infer functional relevance and analyse regulatory mechanisms across diverse datasets. We aim to validate these insights using data from degron-based perturbation experiments. Remarkably, genes ranked by SHAP importance predict direct targets of perturbation even from unperturbed data, enabling inference without costly experimental interventions. Our analysis reveals that SHAP not only predicts differential gene expression but also captures the magnitude of transcriptional changes. We validate the cooperative roles of SET1A and ZC3H4 at promoters and uncover novel regulatory contributions of ZC3H4 at gene bodies in influencing transcription. Cross-dataset validation uncovers unexpected connections between ZC3H4, a component of the Restrictor complex, and INTS11, part of the Integrator complex, suggesting crosstalk mediated by H3K4me3 and the SET1/COMPASS complex in transcriptional regulation. These findings highlight the power of integrating predictive modelling and experimental validation to unravel complex context-dependent regulatory networks and generate novel biological hypotheses.

## Author summary

Genes are turned on or off through complex processes involving many proteins that interact with DNA wrapped histones and modify their structure. These changes, known as epigenetic modifications, help control how genes are expressed without altering the DNA sequence itself. In this study, we wanted to understand how different proteins influence gene activity in mouse stem cells by looking at their positions along the

following accession numbers: GSE199805, GSE181714 and GSE159400. The source code, processed data and model weights are publicly available on the GitHub repository https://github.com/kashyapchhatbar/SHAP-analysis.

**Funding:** KC was supported by a scholarship from College of Science and Engineering, University of Edinburgh. This work was supported by a Wellcome Investigator Award to AB (ref. 222507), a European Research Council Advanced grant (ref. Gen-Epix - 694295) and a core grant (ref. 203149) to the Wellcome Centre for Cell Biology. GS acknowledges co-funding from Next Generation EU, in the context of the National Recovery and Resilience Plan, Investment PE1 - Project FAIR "Future Artificial Intelligence Research". This resource was co-financed by the Next Generation EU (DM 1555 del 11.10.22). The funders had no role in study design, data collection and analysis, decision to publish, or preparation of the manuscript.

**Competing interests:** The authors have declared that no competing interest exists.

genome, particularly whether they act near the gene's start site (promoter) or within the gene body. To do this, we used machine learning models and a method called SHAP, which helps explain the model's decisions. By comparing our predictions to data from experiments where specific proteins were removed, we found that some proteins have context-specific effects, acting not only at the promoter but also along the whole gene body. Our approach highlighted both well-known and unexpected regulators of transcription and revealed that gene body signals, which are often overlooked, can play key roles. These findings show how explainable AI can help uncover new insights into how epigenetic features shape gene regulation, and offer a powerful way to generate testable hypotheses from complex genomic data.

## Introduction

Understanding how cells regulate gene expression during physiological or disease processes remains one of the most important open problems in biology, with potentially immense implications for biomedicine and biotechnology. While the fundamental process of regulation by transcription factor binding has been extensively studied [1], the complexity of sequence-dependent signal integration in a crowded chromatin environment remains largely unknown. Through efforts to tackle this outstanding problem, the last decade has witnessed the development of several large-scale international efforts to profile multiple epigenomic marks across a variety of cellular contexts [2–4]. The large amounts of data made publicly available by these projects raised the prospect of a complementary, data-driven approach to understand general mechanisms by which gene expression is regulated. Numerous studies have used statistical models to predict gene expression or other molecular features from genomic sequence and epigenomic marks [5–11]. More recently, advances in artificial intelligence (AI) using a variety of neural architectures have been brought to bear, allowing increasingly accurate prediction and imputation of gene expression as well as other molecular features [12–15]. A critical limitation of this data-driven approach is its entirely correlative nature. Because the vast majority of data sets are derived from cells at steady-state, every prediction will demand independent validation via explicit perturbation experiments, a requirement that has rarely been met.

Since complex models generally trade off biological interpretability against statistical accuracy, so-called explainable AI (XAI) techniques are adopted to provide human interpretable explanations for specific model predictions and to hypothesise potential underpinning biological mechanisms [16–18]. In this paper, we set out to benchmark the efficacy of XAI approaches in identifying regulatory processes by taking advantage of the recent availability of dynamical measurements provided after rapid perturbations via auxin-inducible degron (AID) and degradation tag (dTAG) systems [19]. We set out to predict RNA Polymerase II occupancy from several chromatin associated proteins and epigenomic marks, quantifying the influence of each protein or mark on each gene using SHAP (Shapley Additive Explanations) [17], one of the most widely used XAI approaches. We then validate the efficacy of the learnt explanations by systematic comparisons of direct target and random gene sets following degradation of one or more factors. Our findings demonstrate the utility of combining predictive modelling with SHAP analysis to uncover complex biological interactions and infer functional mechanisms in transcriptional regulation. By leveraging this integrative approach, we dissected the roles of chromatin-associated proteins, uncovered unexpected functional relationships, and validated the robustness of SHAP across datasets.

## Results

### Data sets used for modelling

To ensure a robust and comprehensive analysis of transcriptional regulation, we utilised datasets from three independent studies spanning different experimental designs and chromatin-associated protein targets: Dobrinić et al. (2021) [20], Hughes et al. (2023) [21], and Wang et al. (2023) [22] (Table 1). Dobrinić and colleagues [20] investigated the roles of PRC1 [23,24] and PRC2 [25,26] complexes in Polycomb-mediated transcriptional repression, using ChIP-seq data for RING1B, SUZ12, and associated histone modifications such as H3K27me3 and H2AK119ub1 to evaluate their contributions. Additionally, this study included degron-based perturbation experiments for RING1B and SUZ12, enabling a direct assessment of acute transcriptional changes using nuclear RNA-seq assays. Hughes and colleagues [21] focused on transcriptional regulation mediated by SET1A [27] and ZC3H4 [28], two chromatin-associated proteins with well-characterised roles in activating and restricting transcription, respectively. This study leveraged ChIP-seq data for SET1A, ZC3H4, and H3K4me3, alongside degron-based perturbation experiments, to uncover the interplay between these factors to regulate transcription. Wang and colleagues [22] explored the role of H3K4me3 in transcription initiation and RNA Pol-II pause-release, utilizing ChIP-seq data for the SET1 complex components DPY30 and RBBP5 [27], Integrator complex member INTS11 [29,30], and factors involved in transcription elongation and Pol-II pause-release such as CDK9 [31], BRD4 [32], HEXIM1, NELFA [33], and AFF4 [34]. Transcriptional dynamics after acute perturbations were assayed using TT-chem-seq [21,22]. Together, these datasets provided a rich resource for examining the relationships between chromatin modifications and transcriptional regulation.

### Modelling of protein occupancy predicts transcription output

In order to understand the relationships between protein occupancy profiles and transcriptional regulation, we built deep neural network (DNN) models, specifically multilayer perceptrons (MLPs), gradient-boosted trees (XGBoost) and linear regression to predict RNA Pol-II occupancy from ChIP-seq data (Fig 1A). MLPs are prioritised in this study for their competitive performance and efficiency in regression tasks [35,36], along with their scalability and potential for improved interpretability using XAI approaches. Since epigenomic data is not necessarily statistically independent, we also utilised gradient-boosted trees. To enhance the resolution of our predictions, normalised ChIP signals were quantified separately from promoter regions and gene body regions, enabling us to investigate the distinct contributions of chromatin proteins and marks to transcription regulation across these regions. By integrating these separate measurements, we aimed to identify specific factors or chromatin features that predominantly influence transcription at promoters or gene bodies. For all data sets, the occupancy of RNA Pol-II was predicted with high precision ($R^2$ ranging between 0.85–0.95) (Fig 1B). Model performance was assessed with shuffled 5-fold KFold cross-validation.

We hypothesise that the model, trained on data assayed from unperturbed conditions, captures biologically meaningful relationships between protein occupancy and transcriptional output. To interpret these relationships, we use SHAP (SHapley Additive exPlanations) [17, 37], an XAI approach that quantifies the contribution of each input feature to the model's predictions while potentially accounting for complex, non-linear interactions (Fig 1A). After we modelled the data using MLP and XGBoost regression, we utilise KernelSHAP, DeepSHAP and TreeSHAP algorithms from the SHAP library. Through comparing SHAP values between direct transcriptional targets (genes that exhibit significant transcriptional changes following

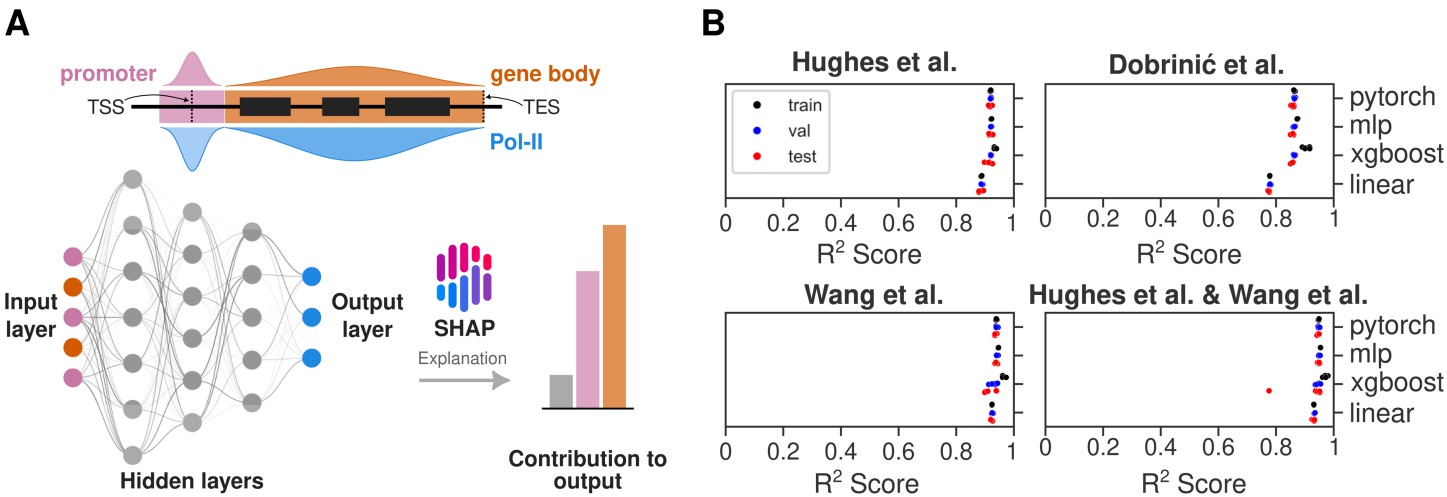

**Fig 1. Overview of predictive modelling for RNA Pol-II occupancy.** (A) Schematic of the multilayer perceptron (MLP) model architecture used to predict RNA Pol-II occupancy based on chromatin-associated protein profiles. Input features include protein occupancy data from ChIP-seq experiments, and outputs represent predicted RNA Pol-II occupancy. (B) Model performance for predicting RNA Pol-II occupancy based on chromatin-associated protein profiles across datasets. Coefficients of determination ($R^2$) are shown for training, validation and test datasets for 5 splits.

rapid protein degradation) and random genes, we aim to assess the relative importance of specific protein occupancy features at promoter and gene body context in driving transcriptional outcomes. For robust comparison, we categorised the genes derived from data generated from nascent (TT-seq) or steady state RNA-seq experiments after acute degradation of the protein of interest. The categorisation was based on the significance threshold using p-adjusted values attributed to the confidence in estimating the fold change. We calculated the mean absolute SHAP values for each group and the error is estimated through random subsampling. Since the model is trained on data assayed before any perturbations, differences in SHAP values between direct targets (p-adjusted value < 0.05) and random genes (p-adjusted value > 0.05) may indicate SHAP's potential to identify functional relevance. After successful training, SHAP values are computed for gene loci present only in the test dataset to prevent data leakage. Because direct targets constitute 6–28% of genes in our degron datasets, every test split was built to preserve that proportion. For each split we drew 50 balanced subsamples, each containing 50 direct-target genes (p-adjusted value < 0.05) and 50 randomly selected non-targets (p-adjusted value > 0.05). Absolute SHAP values were averaged across these repeats to minimise bias from class imbalance.

SHAP value is an estimate of the original Shapley value [37] which is a unique solution that fairly distributes the difference between the model prediction and a background expectation across all features. Because these estimations possess desirable properties of local accuracy, missingness and consistency, summing all values reconstructs the predicted RNA Pol-II occupancy for each gene. These surprising and unique attributes allows direct comparison of feature contributions on a gene-by-gene or gene-set-by-gene-set basis. To benchmark SHAP's effectiveness against traditional approaches, we also conducted separate linear regression analyses for target and random genes. To establish a challenging benchmark, a linear regression baseline was employed, deliberately leveraging outcome-related information that would be inaccessible at inference time for the SHAP-enabled models. Linear regression coefficients can provide only global insights, revealing general trends across the entire set of target or random genes. In contrast, SHAP analysis calculates feature importance scores for each

gene individually, offering unprecedented resolution into how different regulatory features influence transcription in a gene-specific manner. We anticipated that this gene-by-gene analysis would reveal complex, context-dependent interactions between features that would be masked by global linear trends. The comparison of absolute mean values for different gene sets whether they are direct targets or not allows estimating whether a particular mark or multiple marks contribute significantly in predicting RNA Pol-II occupancy. This approach was designed to test whether SHAP's ability to capture gene-specific regulatory patterns could provide deeper mechanistic insights than traditional analyses.

## Strengths and limitations of SHAP analysis in resolving co-occupying transcriptional regulators

To evaluate SHAP's ability to accurately attribute importance to individual proteins in complex regulatory systems, we generated a highly accurate model (Fig 1B) to predict RNA Pol-II occupancy using PRC1, PRC2 and H3K4me3 ChIP-seq data as they were sequenced together in the Dobrinić et al. study [20] (Fig 2A). PRC1, through its core subunit RING1B [23,24], functions as the primary transcriptional repressor [38], while PRC2 reinforces repression by depositing the H3K27me3 histone mark [39]. The extensive co-occupancy of PRC1 and PRC2 at target loci [40] provided an ideal system to test SHAP's performance in disentangling overlapping regulatory inputs. As expected, RING1B promoter occupancy emerged as significantly more important at direct target gene loci (S1A Fig) when SHAP values were calculated using DeepSHAP algorithm. When we extended the analysis using KernelSHAP and TreeSHAP across all time points (2h, 4h, 8h, and 24h; S1B Fig) following RING1B depletion, we consistently identified RING1B as more important at direct targets. This robust finding aligns with extensive research that has established PRC1's dominant role in early transcriptional repression. This consistency represents a significant achievement, as it demonstrates SHAP's ability to capture biologically meaningful features, even in a highly co-linear regulatory environment. Across all time points, SHAP attributed greater importance to SUZ12 and its catalytic histone mark H3K27me3 at direct target genes of RING1B than at random genes, yet both still ranked below RING1B itself (S1B Fig). In all our SHAP explainers except TreeSHAP at 2h and 4h, RING1B was ranked higher than H3K27me3 (S1B Fig). It is important to note that the model was trained on unperturbed data, and thus it cannot reasonably infer or capture the temporal ordering of regulatory effects. The dynamics of feedback mechanism between PRC1 and PRC2 [25,41–43] makes it plausible for H3K27me3, which is deposited by SUZ12 to show higher importance at these time points.

Interestingly, SHAP also highlighted the importance of H3K4me3 at all gene loci regardless of perturbation following RING1B depletion (S1A and S1B Fig), raising the possibility that the SET1 complex, which deposits this histone mark [27] is recruited to polycomb-repressed genes. This observation aligns with the established roles of CFP1, a SET1 complex component that binds CpG islands (CGIs) [44], and KDM2B, a PRC1 complex recruiter which also binds CGIs [45]. These findings provide further evidence of a dynamic interplay between the SET1 and PRC complexes, wherein CFP1 and KDM2B continuously sample CGIs but ultimately yield to the reinforcing feedback mechanisms of PRC1 and PRC2 [46], maintaining transcriptionally repressed loci. These findings also raise the possibility that these loci represent "bivalent" promoters, where H3K4me3 and H3K27me3 coexist [47]. This dual marking is thought to safeguard proper and robust differentiation by maintaining genes in a poised state, balancing activation and repression until lineage-specific cues drive transcriptional commitment [47,48], further emphasizing the dynamic regulatory interplay revealed by our analysis. Another possibility is H3K4me3's well documented association with RNA Pol-II. We reasoned

**A** Cartoon representation of the chromatin-associated proteins and histone marks

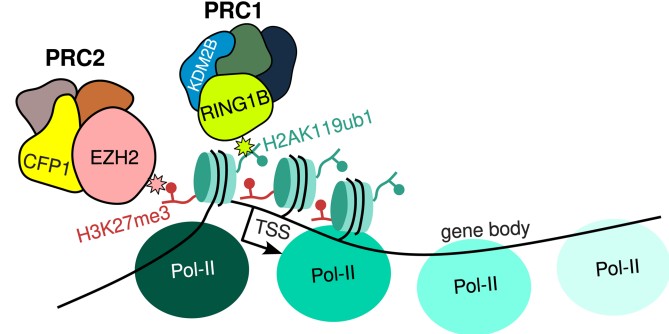

**B** SHAP importance for direct targets identified using RING1B degron (2h)

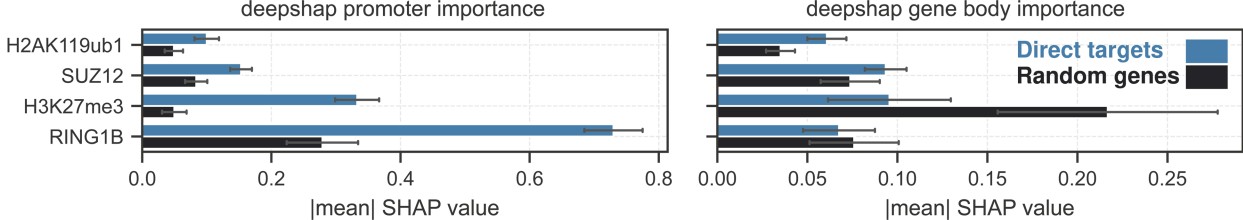

**C** SHAP importance computed across multiple models using multiple algorithms

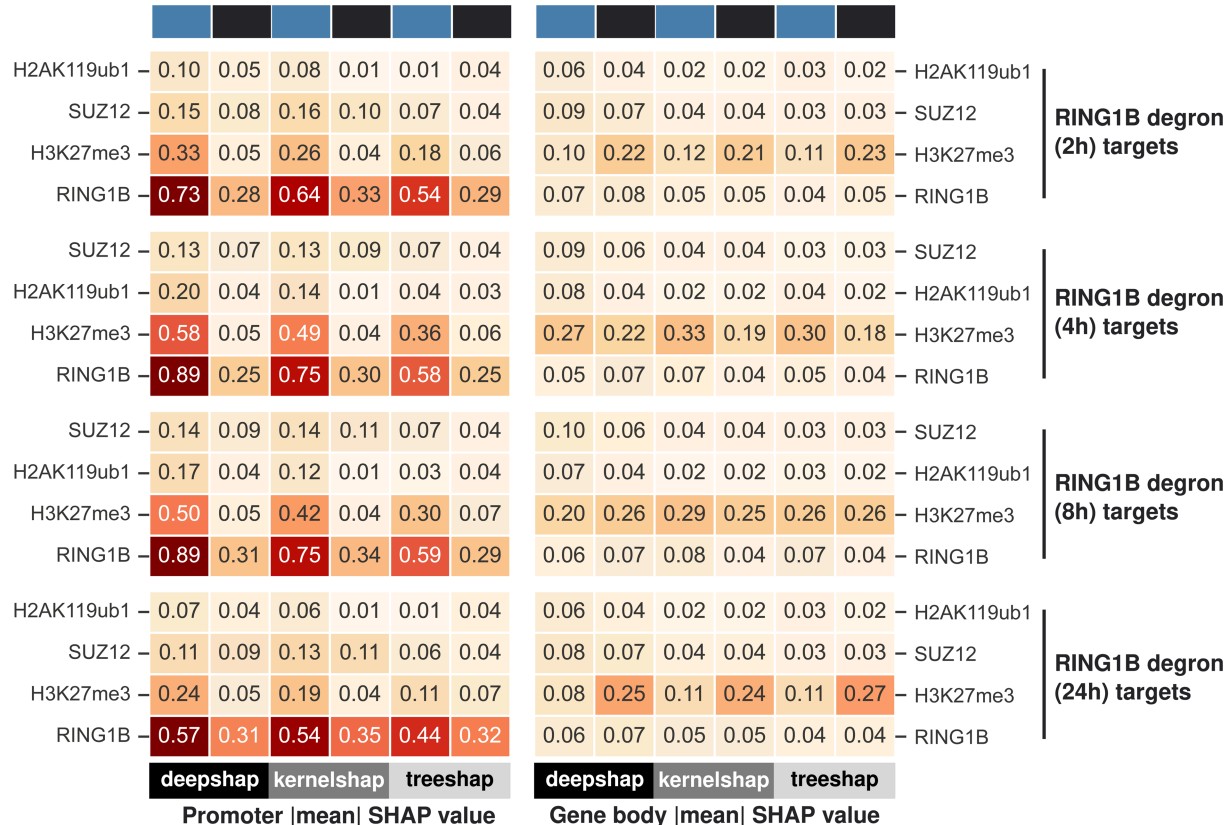

**Fig 2. SHAP values for co-occupying PRC1 (RING1B) and PRC2 (SUZ12) components and their catalytic marks.** (A) Chromatin-associated proteins and histone marks relevant for this model. (B) Comparison of absolute mean DeepSHAP values computed for direct targets (RING1B degron 2h) and random genes at promoter and gene body contexts. Random subsampling is used to calculate intra-category variability and the error bars indicate the standard deviation of one-half. (C) Heatmap of absolute mean SHAP values for different algorithms DeepSHAP, KernelSHAP and TreeSHAP is plotted for direct targets (blue) and random genes (black) across all time points following RING1B degradation (2h, 4h, 8h and 24h).

that this model is relying heavily on H3K4me3 to predict RNA Pol-II occupancy. Therefore, we trained new models without incorporating H3K4me3 occupancy. Surprisingly, these new models retained similar levels of predictive accuracy with test $R^2$ scores dropping from 0.85 (Fig 1B) to 0.8 (S2A Fig) indicating that the modelling strategy does not rely heavily on the correlative relationship between H3K4me3 and RNA Pol-II. When these updated models were interpreted using multiple interpretability algorithms, absolute SHAP values for RING1B promoter occupancy were consistently higher for target gene loci across all time points (Fig 2B, C), following RING1B depletion. Surprisingly, regression coefficients obtained from linear models failed to highlight any significant difference between direct targets and random genes across all time points (S2B, S2C, S2D and S2E Fig). These results demonstrate the utility of SHAP in consistently identifying biologically validated regulators like RING1B as key players, even in complex and co-linear regulatory systems. We argue that our analysis of chromatin-associated proteins, where SHAP values are significantly higher for target genes compared to random genes, is analogous to assessing the functional impact of these proteins.

## Interpretable modelling analysis aids in inferring the combined role of SET1A and ZC3H4 in transcriptional regulation

A recent study [21] revealed the combined role of SET1A and ZC3H4 in transcriptional regulation, highlighting their interplay. Building on these datasets, we aimed to determine whether XAI approaches could infer relationships between these proteins directly from predictive models trained on unperturbed data. Understanding such cooperative roles demands methodologies that go beyond traditional bioinformatic techniques, which often rely on summary statistics and correlation analyses to infer biological mechanisms. For instance, SET1A, a histone methyltransferase responsible for H3K4 trimethylation [27], co-occupies promoters with H3K4me3. SET1A, in complex with WDR82, functions as a transcriptional activator, while ZC3H4, also interacting with WDR82, is part of the restrictor complex that regulates non-coding transcription [28,49].

To shed light on their roles in transcriptional regulation, we utilised our highly accurate model for Hughes et al. [21] data (Fig 1B) which predicts RNA Pol-II occupancy using the occupancy profiles of SET1A, ZC3H4, and H3K4me3 (Fig 3A). We performed SHAP analysis on gene sets following the acute degradation of SET1A, ZC3H4, and both proteins simultaneously. SHAP values were consistently higher for SET1A, ZC3H4 and H3K4me3 promoter occupancy in direct targets compared to random genes (Fig 3B), indicating that the model captures biologically relevant effects of these proteins on transcriptional regulation. Additionally, SHAP analysis highlighted a higher importance of ZC3H4 gene body occupancy in target genes, whereas SET1A showed no significant increase in gene body importance (Fig 3B). As a control comparison, we trained separate linear regression models on direct targets and random genes to evaluate whether the relationships between input features and transcriptional regulation differ across conditions. The coefficients derived from these models showed no significant differences (S3C Fig), indicating that the predictive relationships remain consistent between direct targets and random gene sets. These results demonstrate that XAI approaches, such as SHAP, provide valuable insights into the cooperative roles of SET1A and ZC3H4 at promoters compared to linear regression coefficients. When we extended the analysis using multiple SHAP algorithms and direct targets from ZC3H4 degron and SET1A+ZC3H4 degron, we consistently found SET1A, ZC3H4 and H3K4me3 important at promoter context and ZC3H4 at gene body context (Fig 3C, S3A and S3B Fig). The promoter context finding aligns with established experimental evidence [21], validating the robustness of our approach. Surprisingly, the distinct regulatory role of ZC3H4 at gene body represents a potentially novel

**A** Cartoon representation of the chromatin-associated proteins and histone marks

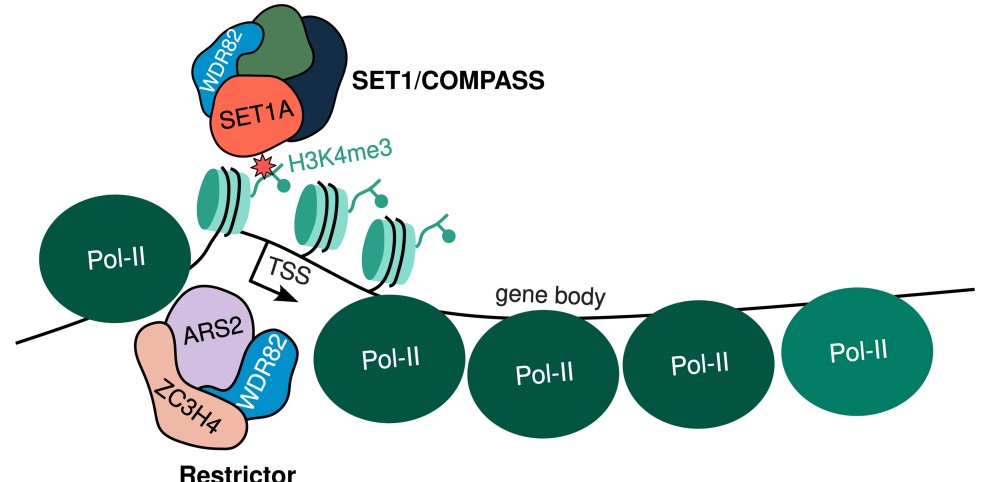

**B** SHAP importance for direct targets identified using SET1A degron

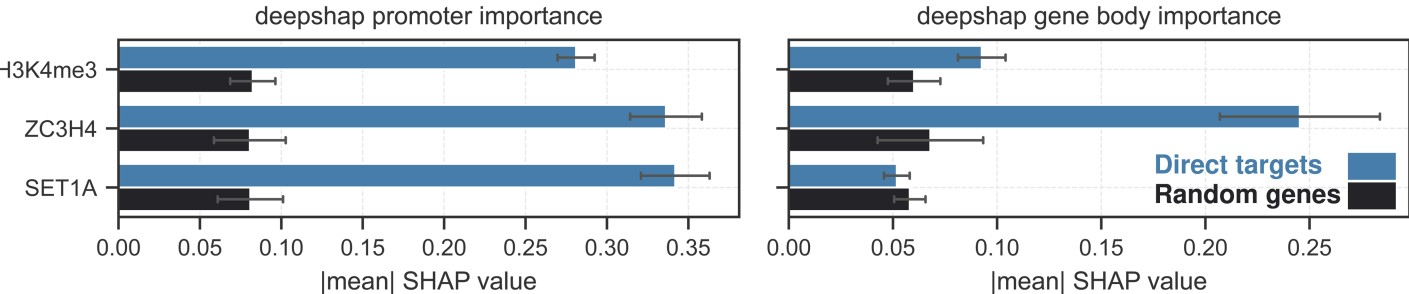

**C** SHAP importance computed across multiple models using multiple algorithms

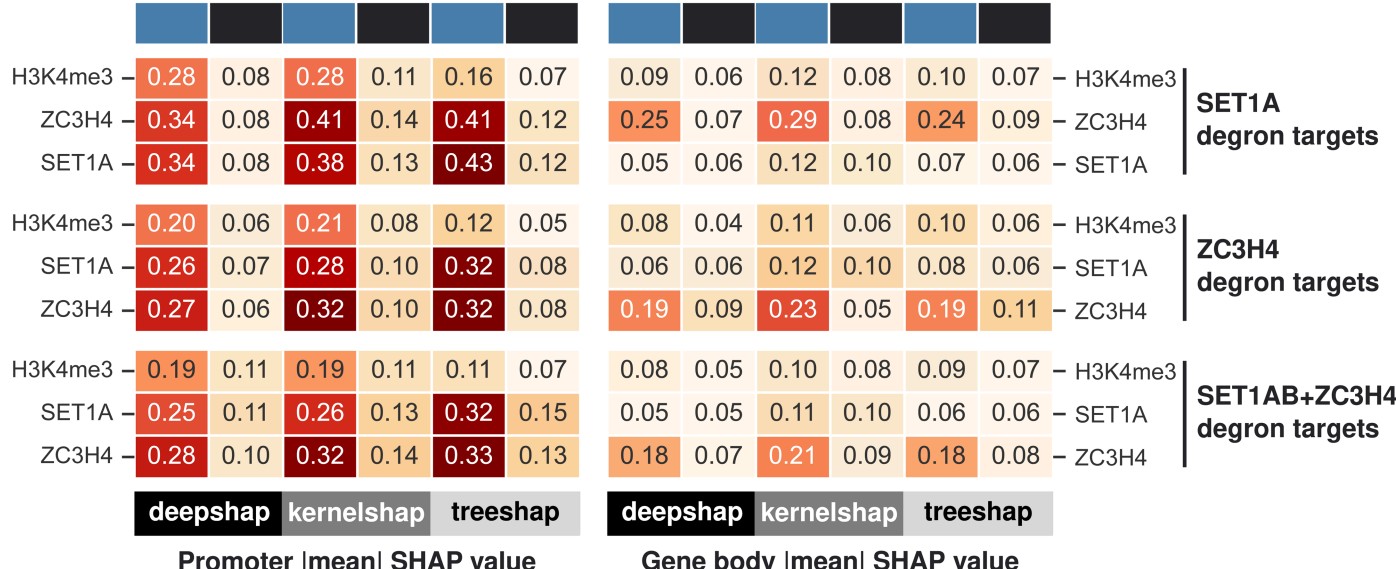

**Fig 3. SHAP values for SET1A, H3K4me3 and ZC3H4.** (A) Chromatin-associated proteins and histone marks relevant for this model. (B) Comparison of absolute mean DeepSHAP values computed for direct targets (SET1A degron) and random genes at promoter and gene body contexts. Random subsampling is used to calculate intra-category variability and the error bars indicate the standard deviation of one-half. (C) Heatmap of absolute mean SHAP values for different algorithms DeepSHAP, KernelSHAP and TreeSHAP is plotted for direct targets (blue) and random genes (black) following SET1A degradation, ZC3H4 degradation and SET1A+ZC3H4 degradation simultaneously.

phenomenon, extending beyond published conclusions and uncovering novel aspects of ZC3H4's contribution to transcriptional regulation.

## SHAP analysis validates the importance of INTS11 at target genes

INTS11, a core component of the Integrator complex [29], is recruited to active promoters through H3K4me3 [22]. This interaction highlights the crosstalk between the SET1/COMPASS complex and Integrator complex in regulating transcriptional activity (Fig 4A). To further investigate these relationships, we utilised our model, employing SHAP to quantify the contributions of DPY30, RBBP5 and INTS11 along with CDK9, HEXIM1, BRD4, NELFA, and AFF4, each playing crucial roles in transcription elongation and Pol-II pause-release [31,32,34,50]. When we utilised our model trained on data from Wang et al. [22] (Fig 1B), our DeepSHAP analysis showed that INTS11 and H3K4me3 promoter occupancy exhibited high SHAP values at target gene loci, regardless of whether INTS11 (Fig 4B), DPY30 (S4A Fig), or RBBP5 (S4B Fig) was degraded. This finding underscores INTS11's role, likely facilitated through its interaction with H3K4me3 at active promoters. Notably, H3K4me3 promoter occupancy was also identified as significant contributors to RNA Pol-II occupancy, particularly at INTS11 target gene loci (Fig 4B). Similar to our previous observations, coefficients derived from linear regression models showed little differences (S4C Fig) between models trained separately for direct targets (INTS11 degron) and random genes whereas SHAP values computed from multiple algorithms clearly separated target genes from random genes (Fig 4B, C). Despite DPY30 and RBBP5 being part of the SET1/COMPASS complex depositing H3K4me3, SHAP analysis revealed a lack of significant importance for DPY30 and RBBP5 (Figs 4B, S4A and S4B), an observation that stands in contrast to expectations. We also noticed that INTS11 SHAP values for direct targets of DPY30 degron (S4A Fig) and RBBP5 degron (S4B Fig) showed only marginal increase compared to random genes. Nonetheless, our analysis highlights the importance of H3K4me3 promoter occupancy in the regulation of INTS11 target genes (Fig 4B, C), supporting its role in regulating RNA Pol-II pausing [22]. Our results independently support the role of INTS11 in regulating transcription at target loci, emphasizing its critical function in transcriptional regulation through H3K4me3 recruitment.

## Cross-dataset validation of SHAP analysis suggests interplay between ZC3H4 and INTS11

To investigate the generalizability of our findings across datasets, we performed a cross-dataset comparison by analysing gene sets derived from the degradation of INTS11 (Wang et al. [22]) alongside predictive models trained on independent ChIP-seq data from Hughes et al. [21]. We also conducted the reverse analysis, using gene sets identified as likely direct targets of SET1A, ZC3H4, and the simultaneous degradation of SET1A and ZC3H4 (Hughes et al. [21]) to validate predictive models trained on ChIP-seq data from Wang et al. [22]. This bipartite approach enabled us to evaluate whether SHAP values derived from one dataset could reliably capture the transcriptional regulatory roles in an independent dataset, providing a stringent test of both the predictive model and the SHAP framework. While DPY30 and RBBP5 are core components of the SET1/COMPASS family of complexes [22,27], they are shared across multiple paralogous assemblies [51]. Therefore, we decided to omit DPY30 and RBBP5 degron target genes as they may contain marks deposited by other methyltransferases [52]. Based on the antagonistic relationship between SET1A and ZC3H4 [21] and INTS11's recruitment via H3K4me3 [22], we hypothesise a potential interplay between SET1A, ZC3H4 and INTS11. As predicted, DeepSHAP analysis on

**A** Cartoon representation of chromatin-associated proteins and histone marks

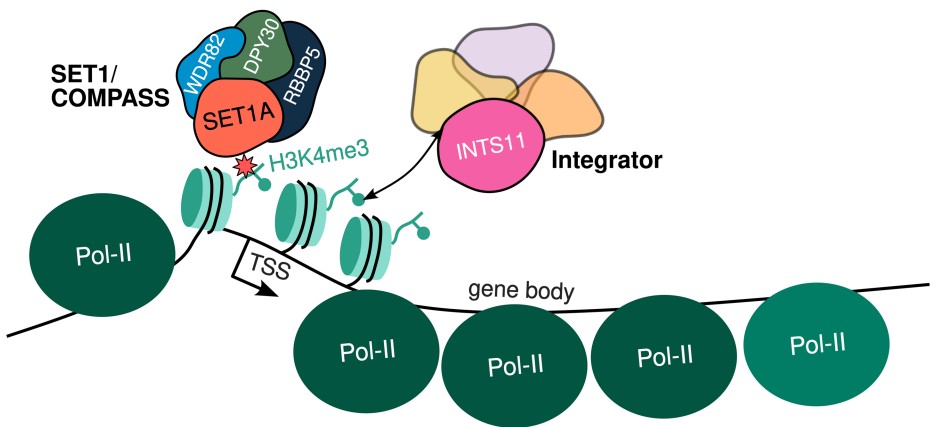

**B** SHAP importance for direct targets identified using INTS11 degron

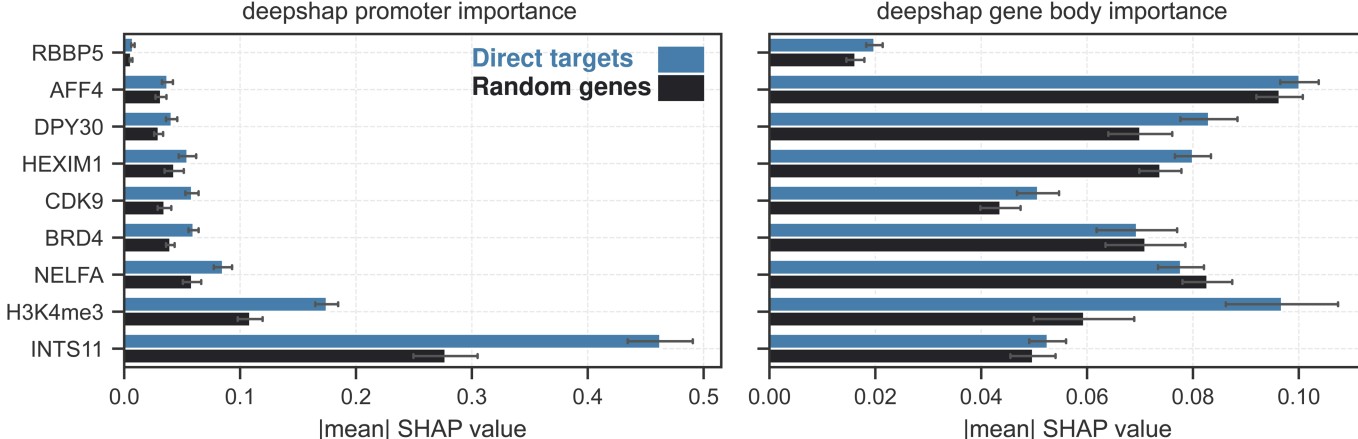

**C** SHAP importance computed across multiple models using multiple algorithms

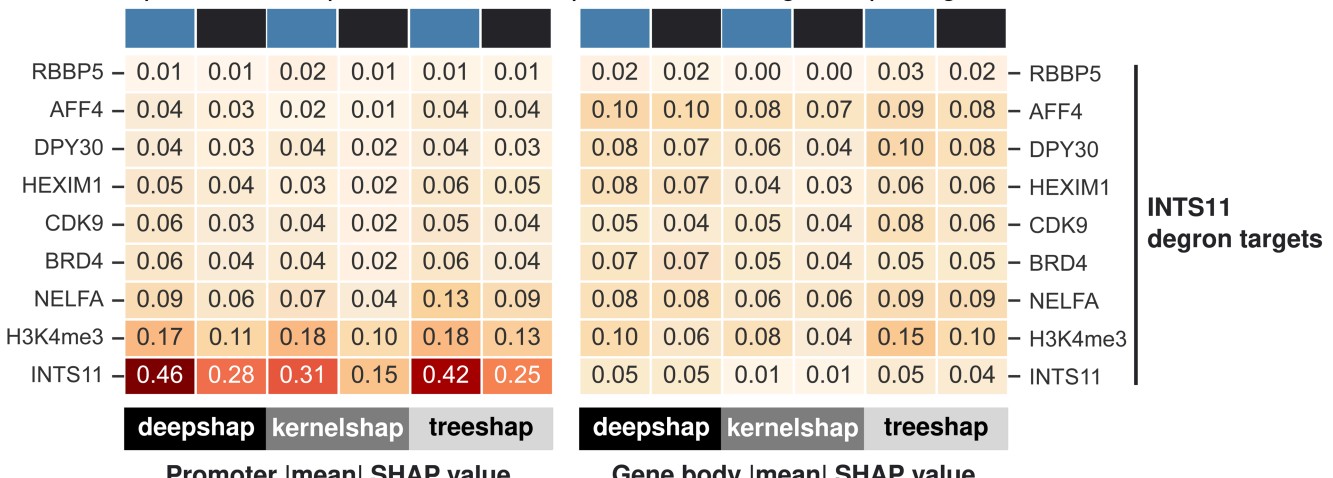

**Fig 4. SHAP values for chromatin-associated proteins profiled in Wang et al. [22] (A) Chromatin-associated proteins and histone marks relevant for this model.** (B) Comparison of absolute mean DeepSHAP values computed for direct targets (INTS11 degron) and random genes at promoter and gene body contexts. Random subsampling is used to calculate intra-category variability and the error bars indicate the standard deviation of one-half. (C) Heatmap of absolute mean SHAP values for different algorithms DeepSHAP, KernelSHAP and TreeSHAP is plotted for direct targets (blue) and random genes (black) following INTS11 degradation.

Hughes et al. model showed higher SHAP values for SET1A and ZC3H4 at INTS11 target gene loci (S5A Fig). When we computed DeepSHAP values for Wang et al. model, the comparative analysis highlighted INTS11 and H3K4me3 as highly important at SET1A target genes (S5B Fig), ZC3H4 target genes (S5C Fig) and SET1A+ZC3H4 target genes (S5D Fig). These observations were consistent when we extended this analysis using multiple SHAP algorithms (Fig 5A).

To probe this potential connection further, we analysed the overlap of target genes between SET1A, INTS11 and ZC3H4. This analysis revealed that SET1A, INTS11 and ZC3H4 share a significant number of target loci (Fig 5B), with highly correlated transcriptional regulatory patterns between INTS11 and ZC3H4 (Fig 5C) suggesting a functional relationship between these two proteins. Reassuringly, we observed anti-correlation between ZC3H4 and SET1A (Fig 5D) transcriptional changes validating the results in Hughes et al. [21]. Surprisingly, transcriptional regulatory patterns anti-correlated between INTS11 and SET1A (Fig 5E), an observation that stands in contrast to observations in Wang et al. [22]. Existing evidence supports the notion that transcriptional machineries often co-occupy the same loci [49], influencing one another's activity. INTS11 is reported to be recruited by H3K4me3 [22] which could explain this unexpected overlap. This reciprocal analysis and significant overlap between target genes of ZC3H4 and INTS11 provides additional support for our Integrator-Restrictor interplay hypothesis.

To test whether SHAP can evaluate a model generated from multiple studies, we combined ChIP-seq data sets from Hughes and Wang studies [21,22] to create a unified data set that includes profiles of key regulatory chromatin proteins and modifications, including SET1A, ZC3H4, INTS11 and H3K4me3 (S7 Fig). Model learned from this unified data set was highly accurate (Fig 1B) and this integrative approach allowed us to assess whether SHAP-derived feature importance remains consistent across multiple studies. DeepSHAP analysis consistently identified ZC3H4, SET1A, INTS11 and H3K4me3 promoter occupancy and ZC3H4 gene body occupancy as most important features for target genes identified from INTS11 degron (S6A Fig), SET1A degron (S6B Fig) and ZC3H4 degron (S6C Fig). When we extended our analysis across multiple SHAP algorithms, we reinforced our findings (Fig 6A). This analysis suggest an unexpected interplay between Integrator and Restrictor complexes and reinforces ZC3H4's dual role in transcriptional regulation, functioning both at promoters and along gene bodies.

## SHAP importance predicts direct targets and correlates with magnitude of differential gene expression

We next sought to determine whether SHAP importance values could better predict transcriptional changes compared to normalised ChIP signals. To evaluate this, genes present in test data were ranked based on their summed SHAP importance values for highly relevant marks (SET1A, INTS11, ZC3H4) and compared to rankings based on normalised ChIP signal of corresponding proteins for which degron experiments are performed. For every ranked gene by SHAP value or normalised ChIP signal, the fraction of target genes within these ranked genes was calculated. Our analysis revealed that genes ranked by SHAP importance consistently exhibited a higher fraction of differentially expressed genes than those ranked by normalised ChIP signals (Fig 6B). This demonstrates that genes identified by ranking of SHAP values are more likely to be direct targets and these insights can be gained without performing degron experiments. Additionally, SHAP values showed stronger correlations with magnitude of differential gene expression compared to correlations with normalised ChIP signal values (Fig 6C). Ranking of absolute mean SHAP values (e.g. Fig 6A) highlight which

**A** SHAP importance computed across multiple models using multiple algorithms

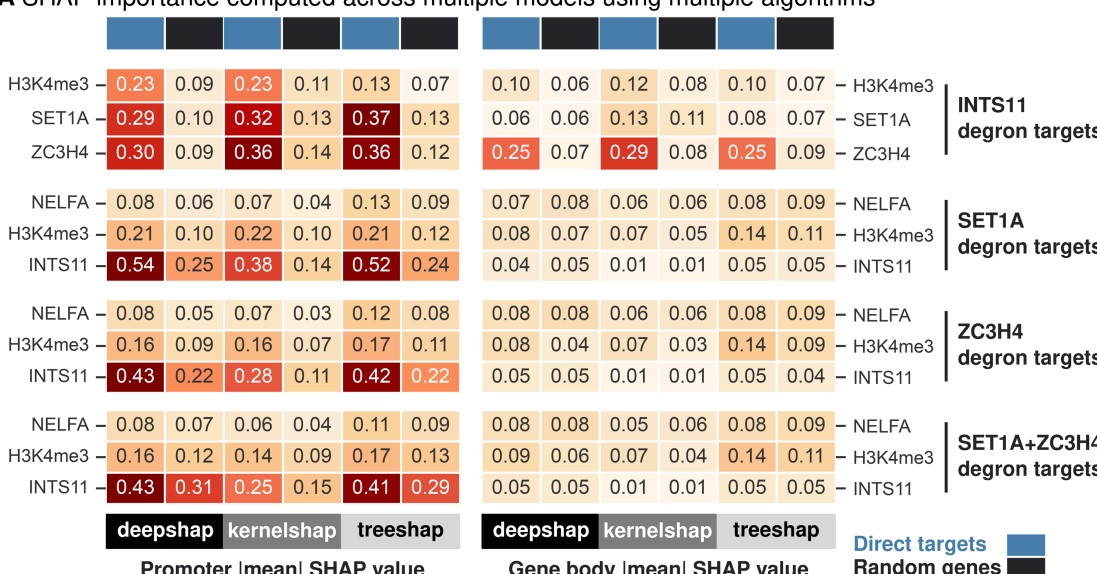

**B** Overlap of direct targets between degrons

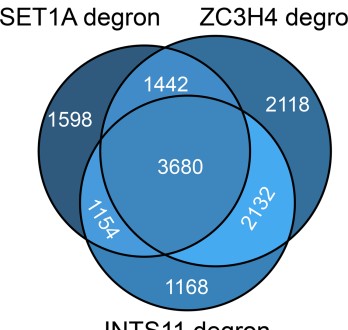

**C** Correlation of the overlapping genes (n=3,680)

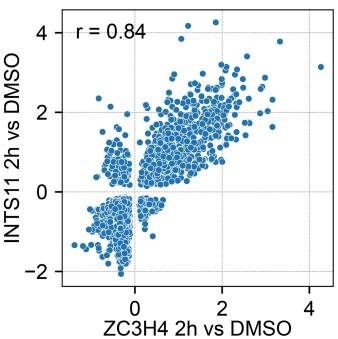

**D** Correlation of the overlapping genes (n=3,680)

**E** Correlation of the overlapping genes (n=3,680)

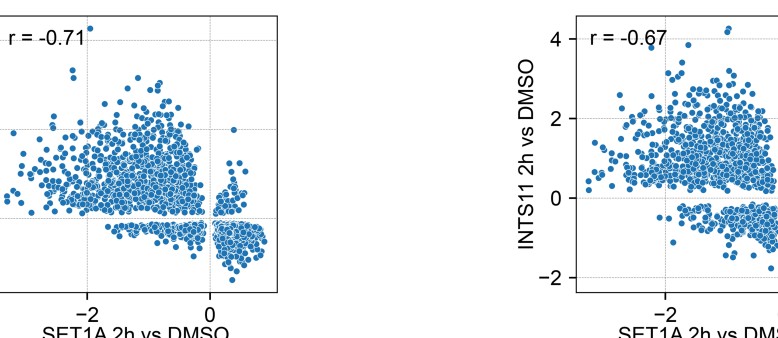

**Fig 5. SHAP values for chromatin-associated proteins profiled in Hughes et al. [21] and Wang et al. [22] (A) Heatmap of absolute mean SHAP values for different algorithms DeepSHAP, KernelSHAP and TreeSHAP is plotted for direct targets (blue) and random genes (black) following INTS11 degradation, SET1A degradation, ZC3H4 degradation and SET1A+ZC3H4 degron targets simultaneously.** (B) Overlap of direct targets of SET1A, ZC3H4 and INTS11. (C) Scatter plot of $log_2$ fold changes following ZC3H4 degradation and INTS11 degradation for the overlapping genes. (D) Scatter plot of $log_2$ fold changes following ZC3H4 degradation and SET1A degradation for the overlapping genes. (E) Scatter plot of $log_2$ fold changes following SET1A degradation and INTS11 degradation for the overlapping genes.

**A** SHAP importance computed across multiple models using multiple algorithms

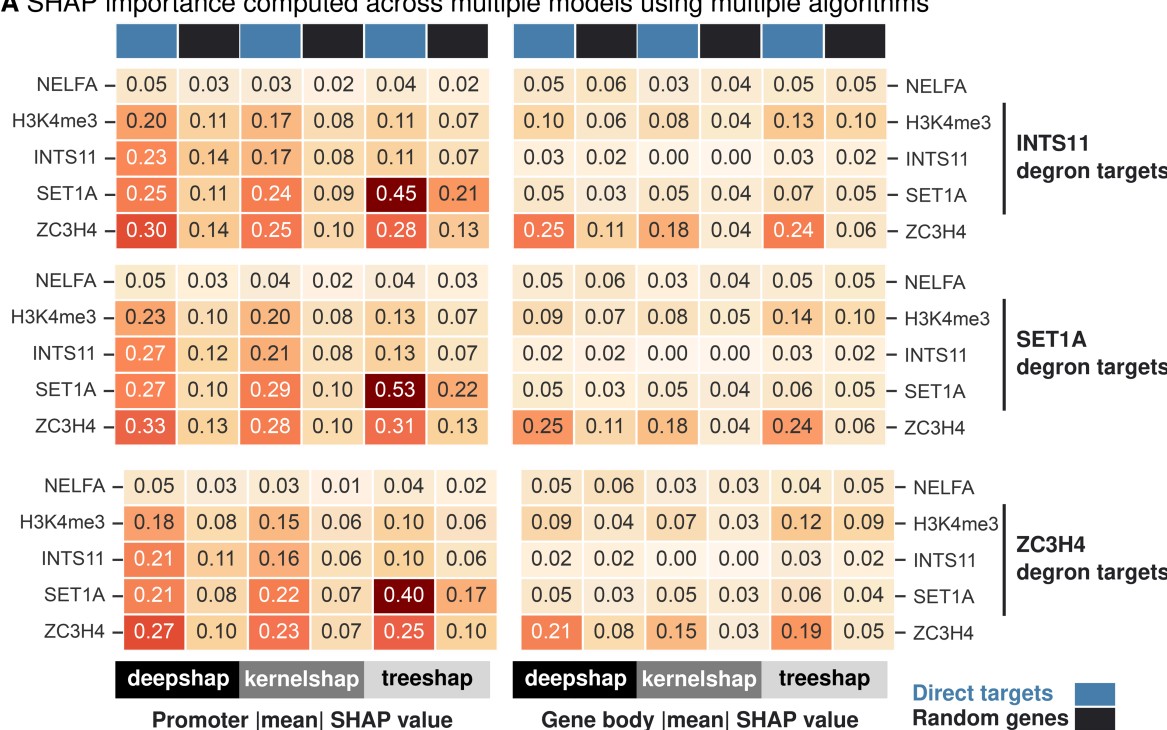

**B** Predictive power of SHAP values to infer direct targets compared to normalized ChIP-seq signal

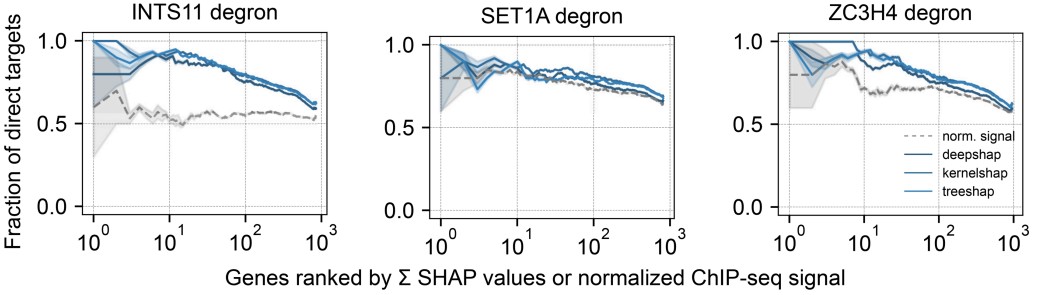

**C** Predictive power of SHAP values to infer fold change of direct targets

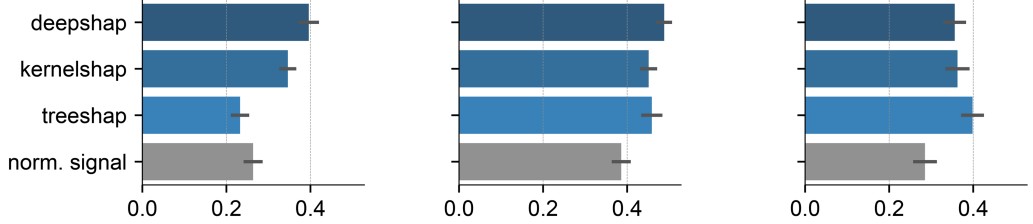

**Fig 6. SHAP values for chromatin-associated proteins profiled in integrative model from Hughes et al. [21] and Wang et al. [22]**
**(A) Heatmap of absolute mean SHAP values for different algorithms DeepSHAP, KernelSHAP and TreeSHAP is plotted for direct targets (blue) and random genes (black) following INTS11 degradation, SET1A degradation and ZC3H4 degradation.** (B) Fraction of overlapping direct target genes with genes ranked by summed SHAP values of INTS11, SET1A and ZC3H4 (blue lines) compared to genes ranked by normalised ChIP-seq signal (grey lines). (C) Absolute pearson correlation between SHAP values from different algorithms and $log_2$ fold change following acute depletion of INTS11, SET1A and ZC3H4. As a control comparison, normalised ChIP-seq was correlated with $log_2$ fold change. Error is estimated by calculating correlation coefficients for all 5 splits and the error bars indicate the standard deviation of one-half.

chromatin-associated proteins to prioritise in future time-resolved degron studies, and the resulting experiments can validate the functional links predicted by our framework. Overall, our analysis highlight the potential of interpretable modelling and the utility of SHAP for data-driven research, hypothesis testing and validation in the genomics field.

## Discussion

A key strength of this study lies in the ability of SHAP analysis to identify biologically meaningful relationships directly from data derived from unperturbed cell systems. The model, trained solely on unperturbed conditions, captures the relationships between protein occupancy and transcriptional output without prior knowledge of perturbations. Remarkably, by ranking genes based on SHAP importance values, direct targets of perturbation can be accurately predicted. This represents a potentially significant achievement, enabling context-dependent inference and complex interplay of regulatory targets without the need for time-consuming and expensive perturbation experiments. We also evaluated differences in SHAP-derived importance for target gene sets derived from perturbation experiments to infer regulatory roles of individual proteins. For instance, the higher SHAP importance of ZC3H4 at promoters and gene bodies of target genes suggests distinct regulatory roles for ZC3H4 and SET1A, highlighting how the model trained on unperturbed systems can reveal insights under perturbed conditions. This observation aligns with recent experimental evidence that ZC3H4 recruits the PNUTS phosphatase [53], and directly controls the phosphorylation of proteins involved in regulating pause-release and elongation [54,55], providing further insight into ZC3H4-mediated transcriptional regulation. In parallel, the consistent importance of INTS11 promoter occupancy highlights its critical role in transcriptional regulation via H3K4me3 recruitment [22]. Cross-dataset validation and integrative modelling reinforced the utility of SHAP-based interpretations, revealing unexpected overlaps between ZC3H4 and INTS11 target loci. This functional connection, supported by shared transcriptional regulatory patterns, raises the intriguing possibility of cooperation between the Restrictor and Integrator complexes (S7 Fig), potentially converging on a common transcriptional termination pathway [49]. Whether the recruitment of INTS11 via H3K4me3 [22] or the antagonistic interplay between ZC3H4 and SET1A [21], which deposits H3K4me3, stimulates a cascade of transcriptional regulation at the same genes remains an open question. Future experiments will be essential to unravel the precise mechanisms underlying these interactions and to determine whether they act sequentially, interdependently, or as part of a larger coordinated regulatory pathway.

The analysis of PRC1 and PRC2 regulatory systems highlights both the strengths and challenges of SHAP in disentangling contributions of co-occupying proteins. SHAP consistently identified RING1B, the core PRC1 subunit, as a key contributor to transcriptional repression, aligning with research that established PRC1's dominant role at early time points [20,38,39]. This result demonstrates SHAP's robustness in capturing biologically meaningful relationships, even in highly co-linear systems. The predominant contribution of EZH2 at some time points signifies the complex interplay between PRC1 and PRC2 [25,41–43]. Notably, SHAP's ability to identify H3K4me3 importance at polycomb-repressed loci suggests potential recruitment of the SET1 complex, uncovering new avenues for understanding the interplay between activation and repression machinery [46]. It could also indicate identification of "bivalent" promoters [47,48], which are a peculiar feature of embryonic stem cells. Overall, these results reaffirm SHAP's utility as a powerful tool for identifying key regulators and generating novel biological hypotheses, while acknowledging the challenges posed by systems with overlapping regulatory inputs.

Several papers have aimed to disentangle the complex network of biochemical signals underpinning the initiation of transcription [56,57]. However, most of these approaches proceeded from a simplifying hypothesis that the expression of different genes are independent processes, conditioned on the local epigenetic environment. Such an assumption necessarily can only infer an average effect of a specific epigenetic factor on gene expression, and requires a very comprehensive assaying of epigenetic marks to be approximately valid. A major benefit of our method is that SHAP values are per gene, and thus directly provide an estimate of context dependence which is increasingly recognised to be a necessary ingredient to understand the mechanisms of gene expression [58]. Our case study highlights the significant advantages of SHAP in providing gene-by-gene explanations for individual predictions, enabling insights into transcriptional regulation that would be difficult to achieve using traditional methods. By integrating predictive modelling with explainable AI, we offer a scalable framework capable of exploring both well-characterised and poorly understood transcriptional processes. A known limitation of SHAP values calculated using the KernelSHAP algorithm is the implicit assumption that input features are statistically independent. For epigenomic data this assumption is rarely met because of the co-localisation of histone modifications, chromatin-associated proteins and RNA Pol-II. We therefore employed other algorithms like TreeSHAP and DeepSHAP which attempt to account for some statistical dependence. We emphasise that SHAP values, regardless of the algorithm used, quantify associations and does not direct causality. We recommend they should always be interpreted in combination with experimental evidence. While the present model centres on promoter- and gene body-occupancies, incorporating regulation through enhancers could broaden its predictive scope, though reliably mapping enhancers to their target genes is an emerging challenge. Overall, this study underscores the transformative potential of combining computational modelling with explainable AI to advance our understanding of transcriptional regulation and generate novel biological hypotheses.

## Materials and methods

### Preprocessing of sequencing data

All bioinformatic analyses were performed using Snakemake workflows [59], ensuring reproducibility and scalability of the analysis pipeline explained in S1 Appendix. These workflows include detailed specifications of software versions, command-line parameters, and dependencies for each analysis step.

**ChIP-seq data analysis.** Raw sequencing data for ChIP-seq experiments were downloaded from the Gene Expression Omnibus (GEO). Reads were aligned to the reference genome using Bowtie2 [60], ensuring high alignment accuracy. To improve data quality, reads mapping to ENCODE blacklisted regions, which are repetitive and highly mappable, were removed. Peak calling and signal track generation were performed with MACS2 [61]. Treatment and control pileup tracks were generated, and the bdgcmp command was used to calculate fold enrichment tracks, normalising ChIP signal to input samples. To quantify the signal, bigWigAverageOverBed [62] was used to calculate the mean signal across defined genomic regions, such as promoters and gene bodies. For each ChIP-seq feature, the average signal intensities were calculated across gene promoters (±1 kb from transcription start site, TSS) and gene bodies (+1kb from TSS to transcription end site, TES). These quantifications were performed using bigWigAverageOverBed, which computes the average signal intensity over specified genomic intervals.

**RNA-seq and TT-seq data analysis.** Raw sequencing data for RNA-seq and TT-seq experiments were also downloaded from GEO. Reads were aligned to the reference

genome using Bowtie2 [60]. PCR duplicates were identified and removed using SAMtools markup [63] to eliminate artifacts due to library preparation. Gene-level read counts were quantified using featureCounts [64] from the Subread package. Differential gene expression analysis was performed using DESeq2 [65] in R, applying default parameters to identify significantly differentially expressed genes.

## Sequencing data sets

Data sets considered for interpretable modelling are listed in Table 1. These data sets include ChIP-seq and RNA-seq data from mouse embryonic stem cells (mESCs) with perturbations targeting chromatin-associated proteins and histone modifications. The data sets were selected based on their relevance to transcriptional regulation and the availability of high-quality sequencing data.

## Data preparation and modelling

To perform SHAP-based model interpretation, we first processed publicly available chromatin profiling ChIP-seq data. Raw sequencing reads and corresponding input controls were downloaded from GEO. All data sources, associated GEO accessions, and chromatin-associated proteins used in this study are listed in Table 1. All reads were processed using a standardised pipeline as described in ChIP-seq data analysis, detailed and its steps visualised in S1 Appendix. The processed data were then used to generate a feature matrix, where each row corresponds to a gene and each column corresponds to a chromatin-associated protein or histone modification. The feature matrix was constructed by averaging the ChIP-seq signal over promoter regions (±1 kb from TSS) and gene bodies (+1 kb from TSS to TES) for each gene.

To predict RNA polymerase II occupancy and assess the contribution of chromatin features, we first generated five distinct train-validation-test (80-10-10%) splits of the dataset. Following this, the data was $log_{10}$ transformed, then centered to zero mean and unit variance. Each of the four model types was independently trained on all five splits to evaluate performance variability and robustness. The models included: (1) a feedforward neural network implemented in PyTorch [66], with hyperparameters (learning rate initiation, weight decay and hidden layer sizes) with ReLu as activation function optimised using Optuna [67]; (2) an MLPRegressor from scikit-learn [68] with hyperparameters (L2 regularization alpha, hidden layer sizes, and a choice of activation function); (3) an XGBoostRegressor from the XGBoost package [69], tuned for learning rate (1e-3 to 1e-1), tree_method ("exact" or "hist"), and max_depth (6 to 18); and (4) a baseline LinearRegression model from scikit-learn [68].

**Table 1. Sequencing data sets considered for interpretable modelling.**

| Accession | Source | Transcription data from degrons (RNA-seq and TT-chem-seq) | Chromatin associated proteins and histone modifications (ChIP-seq) |
|---|---|---|---|
| GSE199805 [21] | mESC | SET1A dTAG, ZC3H4 dTAG, SET1A+ZC3H4 dTAG | SET1A, ZC3H4, H3K4me3, Pol-II |
| GSE181714 [22] | mESC | INTS11 dTAG | INTS11, DPY30, NELFA, H3K4me3, CDK9, AFF4, HEXIM1, BRD4, RBBP5, Pol-II |
| GSE199805 [21], GSE181714 [22] | mESC | INTS11 dTAG, SET1A dTAG, ZC3H4 dTAG, SET1A+ZC3H4 dTAG | INTS11, DPY30, NELFA, H3K4me3, CDK9, AFF4, HEXIM1, BRD4, RBBP5, SET1A, ZC3H4, Pol-II |
| GSE159400 [20] | mESC | RING1B dTAG | RING1B, SUZ12, H2AK119ub1, H3K27me3, H3K4me3, Pol-II |

Before commencing training, the random seed was set to 42. Training, validation and test $R^2$ scores were recorded for all models across the five data splits. Performance was evaluated by plotting $R^2$ values for training, validation, and test sets for both output targets across all configurations, as shown in Fig 1B and S2A Fig. The hyperparameters learned from the training and validation sets are tabulated and the training and validation $R^2$ history for each model across the five splits for every epoch is shown in S1 Appendix.

### Prediction and interpretability analysis

All models were trained to output two distinct values per gene: RNA polymerase II (Pol-II) signal at the promoter and Pol-II signal across the gene body. Accordingly, SHAP values were computed separately for each output using model-appropriate explainers: shap.DeepExplainer for the PyTorch model, shap.KernelExplainer for the MLPRegressor and shap.TreeExplainer for the XGBoostRegressor. For the baseline LinearRegression model, 95% confidence intervals for regression coefficients were calculated. For each gene, SHAP values for promoter features (i.e., chromatin features summarised over promoter regions) were calculated with respect to the Pol-II promoter signal, while SHAP values for gene body features were calculated with respect to the Pol-II gene body signal. This ensured that feature importance was interpreted in the appropriate spatial and functional context for each output.

The background dataset for SHAP analysis was generated using shap.kmeans, excluding any genes identified as direct targets in the corresponding TT-seq perturbation to avoid information leakage. To contextualize SHAP values with transcriptional response, we stratified test-set genes using adjusted p-value thresholds (below and above 0.05) from TT-seq differential expression experiments. For each threshold category, 50 genes were randomly sampled and their absolute mean SHAP values were computed across 50 iterations to evaluate intra-category variability.

We computed pairwise Pearson correlations for every dataset and visualised them as clustered heat-maps, which revealed substantial positive correlations and a clear separation between promoter and gene body signals. Despite this multicollinearity, the mean absolute SHAP scores and the rank order of feature importance were essentially identical across DeepSHAP, KernelSHAP and TreeSHAP (and across train/test splits), demonstrating that our interpretability results are robust to shared signal among features. This analysis is shown in S1 Appendix.

### Software

All modelling analyses, including data preparation, MLP regression [66], XGBoost Regression [69], SHAP analysis [17], and linear regression [68], were performed using Python.

### Supporting information

**S1 Fig. SHAP values for H3K4me3, co-occupying PRC1 (RING1B) and PRC2 (SUZ12) components and their catalytic marks across multiple time points following RING1B degradation.** (A) Absolute mean DeepSHAP values computed for direct targets (RING1B degron 2h) and random genes at promoter and gene body contexts. Random subsampling is used to calculate intra-category variability and the error bars indicate the standard deviation of one-half. (B) Heatmap of absolute mean SHAP values for different algorithms DeepSHAP,

KernelSHAP and TreeSHAP is plotted for direct targets (blue) and random genes (black) across all time points following RING1B degradation (2h, 4h, 8h and 24h).
(TIFF)

**S2 Fig. Model performance and linear regression coefficients for co-occupying PRC1 (RING1B) and PRC2 (SUZ12) components and their catalytic marks across multiple time points following RING1B degradation.** (A) Model performance for predicting RNA Pol-II occupancy based on chromatin-associated protein profiles for Dobrinić et al. without incorporating H3K4me3. Coefficients of determination ($R^2$) are shown for training, validation and test datasets for 5 splits. Coefficients for chromatin-associated proteins and histone marks for direct targets (blue) and random genes (black) for (B) 2h (C) 4h (D) 8h and (E) 24h following RING1B degradation.
(TIFF)

**S3 Fig. SHAP values and linear regression coefficients for SET1A, H3K4me3 and ZC3H4.** Comparison of absolute mean DeepSHAP values computed for (A) direct targets (ZC3H4 degron) and random genes, (B) direct targets (SET1A+ZC3H4 degron) at promoter and gene body contexts. Random subsampling is used to calculate intra-category variability and the error bars indicate the standard deviation of one-half. Coefficients for chromatin-associated proteins and histone marks for direct targets (blue) and random genes (black) for (C) SET1A degron (D) ZC3H4 degron and (E) SET1A+ZC3H4 degron.
(TIFF)

**S4 Fig. SHAP values and linear regression coefficients for chromatin-associated proteins profiled in Wang et al. [22].** Comparison of absolute mean DeepSHAP values computed for (A) direct targets (DPY30 degron) and random genes, (B) direct targets (RBBP5 degron) at promoter and gene body contexts. Random subsampling is used to calculate intra-category variability and the error bars indicate the standard deviation of one-half. Coefficients for chromatin-associated proteins and histone marks for direct targets (blue) and random genes (black) for (C) INTS11 degron.
(TIFF)

**S5 Fig. SHAP values for chromatin-associated proteins profiled in Hughes et al. [21] and Wang et al. [22] Absolute mean DeepSHAP values computed for (A) direct targets (INTS11 degron) (B) direct targets (SET1A degron) (C) direct targets (ZC3H4 degron) (D) direct targets (SET1A+ZC3H4 degron) and random genes at promoter and gene body contexts.** Random subsampling is used to calculate intra-category variability and the error bars indicate the standard deviation of one-half.
(TIFF)

**S6 Fig. SHAP values for chromatin-associated proteins profiled in Hughes et al. [21] and Wang et al. [22] from the integrated model.** Absolute mean DeepSHAP values computed for (A) direct targets (INTS11 degron) (B) direct targets (SET1A degron) (C) direct targets (ZC3H4 degron) and random genes at promoter and gene body contexts. Random subsampling is used to calculate intra-category variability and the error bars indicate the standard deviation of one-half.
(TIFF)

**S7 Fig. Potential crosstalk between Integrator and Restrictor complexes in transcriptional regulation.**
(TIFF)

**S1 Appendix. Supplementary methods.** This document contains a section about interpretable modelling and desirable properties of SHAP algorithms. Additionally, it contains detailed information about the methods used in this study, including data preprocessing, model training, and SHAP analysis.
(PDF)

## Acknowledgments

We are grateful to Sara Giuliani for helpful discussions and feedback on the manuscript.

## Author contributions

**Conceptualization:** Kashyap Chhatbar.

**Data curation:** Kashyap Chhatbar.

**Formal analysis:** Kashyap Chhatbar.

**Funding acquisition:** Adrian Bird.

**Investigation:** Kashyap Chhatbar, Guido Sanguinetti.

**Methodology:** Kashyap Chhatbar.

**Project administration:** Guido Sanguinetti.

**Resources:** Kashyap Chhatbar.

**Software:** Kashyap Chhatbar.

**Supervision:** Adrian Bird, Guido Sanguinetti.

**Validation:** Kashyap Chhatbar.

**Visualization:** Kashyap Chhatbar.

**Writing – original draft:** Kashyap Chhatbar, Guido Sanguinetti.

**Writing – review & editing:** Kashyap Chhatbar, Guido Sanguinetti.

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
