## [Decision Letter · Decision Letter 0]

15 May 2025

PGENETICS-D-25-00135

Unravelling epigenetic regulation of gene expression with explainable AI - a case study leveraging degron data

PLOS Genetics

Dear Dr. Chhatbar,

Thank you for submitting your manuscript to PLOS Genetics. After careful consideration, we feel that it has merit but does not fully meet PLOS Genetics's publication criteria as it currently stands. Therefore, we invite you to submit a revised version of the manuscript that addresses the points raised during the review process.

Please submit your revised manuscript within 60 days Jul 14 2025 11:59PM. If you will need more time than this to complete your revisions, please reply to this message or contact the journal office at plosgenetics@plos.org. Please include the following items when submitting your revised manuscript:

We look forward to receiving your revised manuscript.

Kind regards,

Charles G. Danko, Ph.D.

Guest Editor

PLOS Genetics

John Greally

Section Editor

PLOS Genetics

Aimée Dudley

Editor-in-Chief

PLOS Genetics

Anne Goriely

Editor-in-Chief

PLOS Genetics

**Additional Editor Comments :**

Our sincere apologies for the delay in assessing your manuscript.

**Journal Requirements:**

At this stage, the following Authors/Authors require contributions: Kashyap Chhatbar, Adrian Bird, and Guido Sanguinetti. Please ensure that the full contributions of each author are acknowledged in the "Add/Edit/Remove Authors" section of our submission form.

The list of CRediT author contributions may be found here: https://journals.plos.org/plosgenetics/s/authorship#loc-author-contributions

https://journals.plos.org/plosgenetics/s/submission-guidelines#loc-parts-of-a-submission

5) We notice that your supplementary Figures are included in the manuscript file. Please remove them and upload them with the file type 'Supporting Information'. Please ensure that each Supporting Information file has a legend listed in the manuscript after the references list.

Potential Copyright Issues:

i) Figure 5. Please confirm whether you drew the images / clip-art within the figure panels by hand. If you did not draw the images, please provide (a) a link to the source of the images or icons and their license / terms of use; or (b) written permission from the copyright holder to publish the images or icons under our CC BY 4.0 license. Alternatively, you may replace the images with open source alternatives. See these open source resources you may use to replace images / clip-art:

7) Thank you for stating "The raw sequencing data used in this study were obtained from the Gene Expression Omnibus (GEO) and analyzed as described." Please note that your Data Availability Statement is currently missing the DOI/accession number of each dataset OR a direct link to access each dataset. If your manuscript is accepted for publication, you will be asked to provide these details on a very short timeline. We therefore suggest that you provide this information now, though we will not hold up the peer review process if you are unable.

8) Please amend your detailed Financial Disclosure statement. This is published with the article. It must therefore be completed in full sentences and contain the exact wording you wish to be published.

3) If any authors received a salary from any of your funders, please state which authors and which funders.

9) Please ensure that the funders and grant numbers match between the Financial Disclosure field and the Funding Information tab in your submission form. Note that the funders must be provided in the same order in both places as well. Currently, the funders are different in both locations.

**Reviewers' comments:**

Reviewer's Responses to Questions

Reviewer #1: Chhatbar et al present an analysis of transcriptional regulation based on ChIP-seq data using methods of explainable AI in combination with perturbation data. The approach consists of learning models of RNA polymerase occupancy at promoters as the prediction target using the binding patterns of chromatin associated factors as features. These models are then interpreted using Shapley additive explanations (SHAP-values). The authors make use of the gene specific feature importance quantified by the SHAP values to prioritize the key regulators of transcription. By comparing the SHAP values between direct targets of specific chromatin regulators and other genes in perturbation experiments the authors attempt to infer causal relationships. This aspect of the study is very noteworthy. There are however a number of issues that need to be be clarified.

The manuscript lacks depth in the description of the methods. For example:

- how are promoter regions defined?

- how exactly was the Pol2 signal per gene quantified?

- Which gene annotations were used?

- What was the architecture of the MLP model?

- Which hyper parameters were searched, which one were the optimal combination?

- how were the direct targets in the perturbation analyses defined?

The manuscript lacks a comparison to other predictive modelling approaches (prediction of Pol2 from chromatin factors) that were carried out before. Notably, XGBoost has been used successfully and there is also a specific implementation of. SHAP for this algorithm.

The manuscript heavily relies on the interpretation of the model. Therefore it is key to evaluate the ability of the model to generalise. The authors should demonstrate the ability of the model to make predictions in cell lines not used for training. How is the prediction performance on other cell lines?

The advantage of using SHAP over other feature importance metrics should be demonstrated more clearly. For example: does it provide better results than using gini or similar metrics in random forest classifiers?

The authors should also compare the their results to literature that uses graphical models (or partial correlations) to distinguish direct and indirect relations between chromatin regulators such as https://doi.org/10.1016/j.celrep.2016.01.008 or https://doi.org/10.1093/nar/gku1234

Figure 4A is not easy to interpret. What do the grey lines represent?

Reviewer #2: In "Unravelling epigenetic regulation of gene expression with explainable AI - a case study leveraging degron data," Chhatbar and colleagues demonstrate how to successfully predict RNA Pol-II occupancy based on binding data of various epigenomic factors.

The authors extensively use Shapley Additive Explanations (SHAP) values to interpret the biological underpinnings of their DNN-based predictions. Before publication, the authors should consider a couple of improvements and additional analyses.

1. Since this study primarily builds on the advantages of SHAP and because the readership of PLOS Genetics may not be familiar with Shapley values, I recommend including a paragraph explaining some of the fundamentals of SHAP theory. As the authors present a case study, such an introduction would be critical for all readers who want to conduct similar analyses. In particular, it would help to safeguard against problematic interpretations.

2. The interpretation of SHAP values can be problematic at times. This is particularly the case when the independence assumption is violated, for example, when using Lundberg and Lee. I understand that the marks investigated in the context of this study are, by design, dependent. While I acknowledge that this issue is inherent to the data, I want to encourage the authors to discuss this problem explicitly and explain how it should be addressed when interpreting the values.

3. There are some questions concerning the SHAP analysis. From the paper, the coalition sizes, i.e., the number of features, were somewhat limited. For instance, the analysis involving only GSE199805 used SET1A, ZC3H4, and H3K4me3 (M=3) to predict Pol-II occupancy. To what extent are these features correlated in the data? Is there a substantial difference between the correlations obtained from gene bodies and promoters?

4. It would be interesting to know how the overall Pol-II occupancy prediction performance changes for subsets of markers, e.g., when only using ZC3H4 and H3K4me3. Or even a single marker.

5. To appreciate the feature importance differences for target and non-target genes, i.e., SHAP's potential to infer causality, it would be critical to know which strategies and thresholds were used to make this distinction: How were the target genes defined precisely? How many targets and non-targets are there for the individual analyses?

6. I understand that the authors' GitHub repo probably contains all the information about the architecture, training procedure, thresholds, etc. Nevertheless, I recommend including essential information, such as the exact training strategy using "Scikit-learn's data-splitting functionality" (p. 11), e.g., to explicitly state how overfitting problems were addressed ...

7. Since MLPs are rather bloated, it would be interesting to see how a decision tree-based approach, e.g., random forest or gradient boosting, would work.

Minor issues:

- I may have missed it, but $|med(SHAP)|$ shown in several figures is not defined in the paper.

- In Figure 1B, the test and train data points are not distinguishable.

**Have all data underlying the figures and results presented in the manuscript been provided?**

Reviewer #1: **No: **Not obvious from the manuscript where the data can be found

Reviewer #2: Yes

PLOS authors have the option to publish the peer review history of their article (what does this mean?). If published, this will include your full peer review and any attached files.

Reviewer #1: No

Reviewer #2: No

**Figure resubmission:**
---

## [Decision Letter · Decision Letter 1]

6 Oct 2025

Dear Dr Chhatbar,

We are pleased to inform you that your manuscript entitled "Modelling transcription with explainable AI uncovers context-specific epigenetic gene regulation at promoters and gene bodies" has been editorially accepted for publication in PLOS Genetics. Congratulations!

Yours sincerely,

Charles G. Danko, Ph.D.

Guest Editor

PLOS Genetics

John Greally

Section Editor

PLOS Genetics

Aimée Dudley

Editor-in-Chief

PLOS Genetics

Anne Goriely

Editor-in-Chief

PLOS Genetics

BlueSky: @plos.bsky.social

Comments from the reviewers (if applicable):

Congratulations!

Reviewer's Responses to Questions

**Comments to the Authors:**

Reviewer #2: The authors have addressed all my questions.

**Have all data underlying the figures and results presented in the manuscript been provided?**

Reviewer #2: None

PLOS authors have the option to publish the peer review history of their article (what does this mean?). If published, this will include your full peer review and any attached files.

Reviewer #2: No

**Data Deposition**

http://datadryad.org/submit?journalID=pgenetics&manu=PGENETICS-D-25-00135R1

**Press Queries**

---

## [Editor Report · Acceptance letter]

PGENETICS-D-25-00135R1

Modelling transcription with explainable AI uncovers context-specific epigenetic gene regulation at promoters and gene bodies

Dear Dr Chhatbar,

We are pleased to inform you that your manuscript entitled "Modelling transcription with explainable AI uncovers context-specific epigenetic gene regulation at promoters and gene bodies" has been formally accepted for publication in PLOS Genetics! Your manuscript is now with our production department and you will be notified of the publication date in due course.

With kind regards,

Anita Estes

PLOS Genetics

On behalf of:
